# Polypeptide Composition and Topology Affect Hydrogelation of Star-Shaped Poly(_L_-lysine)-Based Amphiphilic Copolypeptides

**DOI:** 10.3390/gels7030131

**Published:** 2021-08-30

**Authors:** Thi Ha My Phan, Ching-Chia Huang, Yi-Jen Tsai, Jin-Jia Hu, Jeng-Shiung Jan

**Affiliations:** 1Department of Chemical Engineering, National Cheng Kung University, Tainan 70101, Taiwan; n36087113@gs.ncku.edu.tw (T.H.M.P.); s110076@shsh.tw (C.-C.H.); n36094704@gs.ncku.edu.tw (Y.-J.T.); 2Department of Mechanical Engineering, National Yang Ming Chiao Tung University, Hsinchu 30010, Taiwan; 3Hierarchical Green-Energy Materials (Hi-GEM) Research Center, National Cheng Kung University, Tainan 70101, Taiwan

**Keywords:** hydrogels, polypeptide, self-assembly, chain conformation, polymer topology

## Abstract

In this research, we studied the effect of polypeptide composition and topology on the hydrogelation of star-shaped block copolypeptides based on hydrophilic, coil poly(_L_-lysine)_20_ (*s*-PLL_20_) tethered with a hydrophobic, sheet-like polypeptide segment, which is poly(_L_-phenylalanine) (PPhe), poly(_L_-leucine) (PLeu), poly(_L_-valine) (PVal) or poly(_L_-alanine) (PAla) with a degree of polymerization (DP) about 5. We found that the PPhe, PLeu, and PVal segments are good hydrogelators to promote hydrogelation. The hydrogelation and hydrogel mechanical properties depend on the arm number and hydrophobic polypeptide segment, which are dictated by the amphiphilic balance between polypeptide blocks and the hydrophobic interactions/hydrogen bonding exerted by the hydrophobic polypeptide segment. The star-shaped topology could facilitate their hydrogelation due to the branching chains serving as multiple interacting depots between hydrophobic polypeptide segments. The 6-armed diblock copolypeptides have better hydrogelation ability than 3-armed ones and *s*-PLL-*b*-PPhe exhibits better hydrogelation ability than *s*-PLL-*b*-PVal and *s*-PLL-*b*-PLeu due to the additional cation–π and π–π interactions. This study highlights that polypeptide composition and topology could be additional parameters to manipulate polypeptide hydrogelation.

## 1. Introduction

Hydrogels, which are three-dimensional networks of hydrophilic polymers, are known to contain a large amount of water while maintaining the structures via chemical and physical cross-linking of individual polymer chains. Hydrogels based on natural or synthetic polymers can be yielded via forming non-covalent physical interactions or chemically covalent bonds [1,2,3]. Due to the ability to mimic animal tissues, their applications in biomedical fields have been rapidly increasing for several decades [4,5,6,7]. This has led to a risen request for well-defined hydrogelators with adaptable properties for biomedical applications such as advanced wound healing, drug carriers, tissue engineering scaffolds, and investigating biomechanical functions [8,9,10,11]. Moreover, hydrogels have exhibited numerous characteristics such as adjustability, pellucidity, and porosity/permeability. However, most of the known synthetic hydrogels have inferior mechanical properties, compared to some of the natural tissues such as cartilages, tendons, and ligaments [12,13,14].

According to biocompatibility and chemical versatility, the hydrogels synthesized based on polypeptides are nature-inspired materials that have great potentiality [15,16,17,18,19]. Commonly, the most well-known technique for preparing synthetic polypeptide-based polymers is ring-opening polymerization (ROP), which allows the preparation of homo- or copolypeptides with huge volume and superior efficiency [20,21,22,23]. A wide range of synthetic polypeptides has shown their capability to adopt secondary conformations such as random coils, *β*-sheets, or *α*-helices in an aqueous environment, allowing the adjustment of mechanical properties of synthetic polypeptides. Moreover, polypeptides can expand their applications in the biomimetic field by conjugating to other materials such as proteins or nanoparticles [24]. Additionally, synthetic polypeptides possess many functional groups on the side chain, making it possible to modify pre- and post-polymerization [15,23,25].

The self-assembly of amphiphilic copolymers has always been an interesting topic for researchers because of the interconnection of two or more different polymer segments, allowing for the recognition of new material characteristics and the useful application of assembled morphologies. The influence of the balance between hydrophilic and hydrophobic interactions on polypeptide self-assembled morphology has been actively studied. The effect of specific interactions and secondary conformations on block copolypeptides can also be a factor governing polypeptide self-assembly [11,12,16,19,26,27,28]. Herein, we report the synthesis and hydrogelation of star-shaped diblock copolypeptides based on hydrophilic, coil poly(_L_-lysine) (*s*-PLL) blocked with a hydrophobic, sheet-like polypeptide segment, anticipating the hydrogelation. The research progresses to synthesize star-shaped diblock copolypeptides, which have attracted much attention because of their outstanding advantages. A few studies have reported that the star-shaped diblock copolypeptides, which are traditionally synthesized by using dendrimers bearing amino groups as the initiators, could form hydrogels in an aqueous solution and the polypeptide topology could be an additional parameter to manipulate their hydrogelation [29,30,31]. However, the synthesis of dendrimers required numerous and complicated steps. To limit this drawback, we previously applied ROP to synthesize polypeptides using readily available alcohols containing different numbers of OH group as the initiators and the 1,1,3,3-tetramethylguanidine (TMG) as the promoter [28,31,32]. Our previous report demonstrated that the star-shaped PLL copolypeptides grafted with phenyl or indole group could form hydrogels in an aqueous environment [27]. In this study, we turn to a different aspect and study the hydrogelation of star-shaped diblock copolypeptides. Poly(_L_-phenylalanine) (PPhe), poly(_L_-alanine) (PAla), poly(_L_-leucine) (PLeu), and poly(_L_-valine) (PVal) are chosen to be the second block of the star-shaped diblock copolypeptides. It is anticipated that the self-assembly of star-shaped diblock copolypeptides would be different from that of star-shaped graft ones, which could have a direct impact on their hydrogelation. Furthermore, varying the polypeptide composition including arm number and monomer would affect the hydrogelation and gel mechanical strength and the molecular assembly [33,34]. In previous studies, researchers reported that the increase of block copolypeptides arm length and arm number led to decreasing in critical gelation concentration (CGC) and increasing in hydrogelation of polymers [29,31], while it had no obvious trend for graft copolypeptides [28]. Moreover, compared to the linear counterparts, the star-shaped ones could create effective inter/intramolecular interactions between the hydrophobic moieties [35]. It is anticipated that polypeptide composition and topology could be additional parameters to manipulate polypeptide hydrogelation.

## 2. Results and Discussion

### 2.1. Synthesis and Characterization of Polypeptides

Star-shaped diblock copolypeptides based on PZLL tethered with four different polypeptide segments (*s*-PZLL-*b*-PY) were synthesized by sequential ROP of respective NCAs using 3-armed and 6-armed initiators, which are 1,1,1-tri(hydroxymethyl)propane and dipentaerythritol. The synthesis procedures and polypeptide chemical structures are shown in Figure 1. The block ratio, the number-average weight (*M_n_*), and molecular weight distribution (*M_w_*/*M_n_*), based on gel permeation chromatography–light scattering (GPC-LS) and proton nuclear magnetic resonance (^1^H NMR) (Appendix A), were calculated, as shown in Appendix A. As the *s*-PZLL first blocks were prepared, a small portion was drawn from the reaction solution and purified for GPC-LS analysis. Based on ^1^H NMR analysis, the presence of the chemical shifts for all the protons on the initiators and polypeptides confirmed the successful synthesis of the *s*-PZLL homopolypeptides and *s*-PZLL-*b*-PY diblock copolypeptides (Figure 1a,b). The block ratio of initiator to ZLL was determined based on the ratio of the integrated areas for the methylene protons (−C(CH_2_O−)_n_) on the initiator and the benzyl protons (−OCH_2_C_6_H_5_) on *s*-PZLL (Figure 1a). From ^1^H NMR analysis, the calculated degrees of polymerization (DPs) reached a good agreement with the feed molar ratios of the initiator and ZLL NCA (Appendix A). The DPs, which were calculated from ^1^H NMR spectra also received a good agreement with the molecular weights of the *s*-PZLL determined based on the GPC analysis (Appendix A). Based on the ^1^H NMR spectra (for example, Figure 1b), the integral ratios of the protons of the benzyl group on the polypeptide chain (−COOCH_2_C_6_H5 and −CH_2_C_6_H_5_), the *α* protons on *s*-PZLL (or –(CH-(CH_2_)-)-, and the δ and ε protons on the PLeu (-CH(CH_3_)_2_) provided block ratios for both blocks, as summarized in Appendix A. To remove the Z-group, *s*-PZLL-*b*-PY diblock copolypeptides were reacted with HBr for 30 min to obtain *s*-PLL-*b*-PY; confirmed by ^1^H NMR analysis, the percentages of the residual Z group were well below 5% (Figure 1c and Appendix A), consistent with a previous study [30]. It is worth noting that the deprotection step would not result in hydrolysis of ester groups and apparent degradation of star-shaped polypeptides based on our previous studies [28,30,32].

### 2.2. Hydrogelation of Block Polypeptides

At room temperature (RT), *s*-PLL-*b*-PY block copolypeptides were dissolved in deionized (DI) water by using ultrasound and vortex mixer. After placing on a counter overnight, the polypeptides can form transparent hydrogels by dispersing in DI water. The CGCs of the polypeptide hydrogels were determined by the tube inverting method, which ranged from 1.0 to 7.0 wt% (Figure 1 and Table 1). Rather, it is found that 3*s*-PLL_22_-*b*-PAla_5.5_ and 6*s*-PLL_21_-*b*-PAla_5.3_ diblock copolypeptides formed hydrogels at much higher concentrations, which were 22.0 and 15.0 wt%, respectively. Table 1 showed that the CGC values of the hydrogel samples was in the following order, 6*s*-PLL_21_-*b*-PPhe_5.3_ < 6*s*-PLL_21_-*b*-PVal_5_ ~ 6*s*-PLL_21_-*b*-PLeu_4.4_ < 3*s*-PLL_22_-*b*-PPhe_6.3_ < 3*s*-PLL_22_-*b*-PVal_5.1_ < 3*s*-PLL_22_-*b*-PLeu_5.5_. It is obvious that the CGCs were dependent on the arm number and composition. It can be seen that the CGCs of 6-armed polypeptide hydrogels were lower than those of 3-armed ones. It can be attributed that the 6-armed polypeptides exhibited more branching chains than the 3-armed ones. The hydrophobic and hydrophilic balance between polypeptide blocks and hydrogen bonding/hydrophobic interactions exerted by the PY segments affected the hydrogelation. *s*-PLL-*b*-PPhe exhibited lower CGC than *s*-PLL-*b*-PVal and *s*-PLL-*b*-PLeu, revealing that the PPhe segment exhibited better hydrogelation ability than the other two segments. It could be attributed that the PPhe segment exhibited additional π–π interactions between benzyl groups and cation–π interactions between PLL and PPhe segments [30,31,36,37].

### 2.3. Molecular Structure of Polypeptide Hydrogels

At neutral conditions, the chain conformations of the star-shaped polypeptides were characterized by circular dichroism (CD) and Fourier transform infrared spectroscopy (FTIR) analyses. The software BeStSel was employed for fitting the CD spectra to compute the secondary conformations adopted by polypeptide chains. The CD analysis revealed that the star-shaped polypeptides adopted mainly random coil and *β*-sheet/turn conformations at neutral conditions (Table 1 and Figure 2a). In agreement with the results determined from the CD analysis, the FTIR spectra of polypeptides exhibited the characteristics of the random coil (1650 cm^−1^), *β*-sheet (1626 cm^−1^), and *β*-turn (1677 cm^−1^) conformations (Figure 2b). Obviously, the results demonstrated that the chain conformation depended on arm number and the second block of the copolypeptides. It is well known that a random coil conformation would be adopted by the PLL segment at neutral conditions. Evidently, the mole fraction of the second segment was lower than the percentage of *β*-sheet/turn conformation. Moreover, 3-armed and 6-armed diblock copolypeptides adopted comparable percentages of both coil and *β*-sheet/turn conformations. It demonstrated that PLL segments, due to their confinement, can adopt more ordered conformation by conjugating with a functional moiety on the side chain or a hydrophobic polypeptide on the chain end [38,39,40,41,42]. Rather, the PLL conformation exhibited little dependence on the conjugation of hydrophilic polymer on a PLL segment [16,43]. It is worth noting that a low percentage of *α*-helical conformation adopted by *s*-PLL-*b*-PLeu was detected by CD (Table 1). Previous studies have shown that a hydrophobic PLeu segment with DP higher than 20 would adopt mainly *α*-helical conformation [44,45].

### 2.4. Morphology and Mechanical Properties of Polypeptide Hydrogels

To characterize the morphology of hydrogels by scanning electron microscope (SEM), the hydrogel samples were frozen in liquid nitrogen and lyophilized to obtain freeze-dried samples. The native morphology of samples observed in the free-dried state was demonstrated in the previous studies [13,14,16,28]. In Figure 3, all the samples exhibited three-dimensional morphology and porous structure. At the same concentration (5 wt%), 6*s*-PLL_21_-*b*-PPhe_5.3_ had a thin elongated fibrillar structure, while 6-armed samples of PLeu and PVal had membranous networks (Figure 3b,d,f). Moreover, unlike the 6*s*-PLL_21_-*b*-PPhe_6.3_ sample, 3*s*-PLL_22_-*b*-PPhe_6.3_ showed membranous morphology. To understand the effect of the arm number and second block on the mechanical strengths of the polypeptide hydrogels, an oscillatory shear rheometer was used to analyze the as-prepared hydrogels formed by *s*-PLL-*b*-PPhe, *s*-PLL-*b*-Pleu, and *s*-PLL-*b*-PVal by conducting angular frequency and strain sweeps (Figure 4). The loss modulus G′′ was obviously smaller than the storage modulus G′. Based on the strain sweep test, it showed that the shear-thinning properties were exhibited on all samples, and the gel network was disrupted at higher strain. Some samples exhibited the shear-thickening property in the loss modulus G′′, while this phenomenon was not seen in the storage modulus G′, which was the same case witnessed in the previous studies [16,28]. The explanation for the shear-thickening mechanism is still a debatable matter. The increase in loss modulus as the storage modulus decreases at high strain might also reflect the increasing virous and liquid-like character of the hydrogels. It can be possibly attributed to the change of the gel network or the formation of force chains [46]. The mechanical strength of the polypeptide hydrogels with 5.0 wt% of polypeptide concentration were arranged in this descending order, 6*s*-PLL_21_-*b*-PPhe_5.3_ > 3*s*-PLL_22_-*b*-PPhe_6.3_ > 6*s*-PLL_21_-*b*-PLeu_4.4_ > 6*s*-PLL_21_-*b*-PVal_5_. At a given polypeptide concentration, the 6-armed polypeptide hydrogels exhibited less rigid and higher mechanical strengths than the 3-armed ones and 6*s*-PLL_21_-*b*-PPhe_5.3_ exhibited the highest mechanical strength among all. It can be seen that the mechanical strength of 3*s*-PLL_22_-*b*-PPhe_6.3_ hydrogel was higher than those of 6*s*-PLL_21_-*b*-PLeu_4.4_ and 6*s*-PLL_21_-*b*-PVal_5_ ones. The 3*s*-PLL_22_-*b*-PLeu_5.5_ hydrogel exhibited comparable mechanical strength with the 3*s*-PLL_22_-*b*-PVal_5.1_ one. The results revealed that the hydrogel rigidity and mechanical strength would depend on the polypeptide composition and topology. Upon large-amplitude strain oscillations, the mechanical strength recovery of these hydrogels was monitored for the investigation of the gel recovery. All the samples demonstrated their ability to progressively recover more than 98% of their original strength after the removal of shearing (Figure 5). The data showed that these star-shaped polypeptide hydrogels have the capability to recover from sol to gel.

### 2.5. Molecular Assembly of Polypeptide Hydrogels

X-ray diffraction analysis (XRD) analysis was employed to determine the macromolecular structure of the freeze-dried hydrogel samples. The presence of a broad peak at 2θ > 20° in the freeze-dried samples demonstrated the presence of an amorphous phase in the samples (Appendix A). Small-angle X-ray scattering (SAXS) analysis was employed to elucidate the packing morphology of the sol and gel samples in the nanometer scale. From SAXS profiles, the region 0.02 Å^−1^ < q < 0.07 Å^−1^ was selected to decide the slope and shape factor. Upon the sol-to-gel transition, all the samples except 3*s*-PLL_22_-*b*-PLeu_5.5_ exhibited the slope (n) transformed from −2 to almost −3.5 in the scattering intensity *I*(*q*) ∝ *q*^n^, while the 3*s*-PLL_22_-*b*-PLeu_5.5_ sample exhibited the slope (n) transformed from −1 to almost −3.5 (Figure 6 and Appendix A). The results suggested all the hydrogel samples self-assembled to form ill-defined three-dimensional (3D) nano-assemblies. In the sol state, all the samples except 3*s*-PLL_22_-*b*-PLeu_5.5_ formed two-dimensional (2D) nano-assemblies [47,48]. Rather, 3*s*-PLL_22_-*b*-PLeu_5.5_ self-assembled to form one-dimensional (1D) fibrillar morphology, analogous to the study by Pine et al., showing that of the linear PLL-*b*-PLeu polypeptide hydrogels comprised of 1D twisted fibrils [33]. In order to quantify the nano-assemblies, SASview software was used to fit the SAXS profiles of 6*s*-PLL-*b*-PY hydrogel samples in the region 0.02 Å^−1^ < q < 0.2 Å^−1^ (Appendix A). The preliminary results showed that the radii of gyration (R_g_) of the nano-assemblies in the 6*s*-PLL_21_-*b*-PPhe_5.3_, 6*s*-PLL_21_-*b*-PLeu_4.4_, and 6*s*-PLL_21_-*b*-PVal_5_ hydrogels were calculated to be 58.5, 97.5, and 47.6 Å, respectively (Table 2). The SAXS profiles of 3*s*-PLL-*b*-PY hydrogel samples cannot be fitted by SASview software due to the presence of a peak in the region 0.02 Å^−1^ < q < 0.03 Å^−1^, which can be used to calculate the characteristic d spacing. The d spacing values for 3*s*-PLL_22_-*b*-PVal_5.1_, 3*s*-PLL_22_-*b*-PPhe_6.3_, and 3*s*-PLL_22_-*b*-PLeu_5.5_ hydrogels were 314, 273, and 241.5 Å, respectively (Appendix A), which could possibly be the distance between the nano-assemblies or other ordered spacing.

### 2.6. Gelation Mechanism of Polypeptide Hydrogels

The above results implied that the hydrophobic, sheet-like PPhe, PLeu, and PVal segments are good hydrogelators to promote hydrogelation of star-shaped PLL-*b*-PY diblock copolypeptides. The hydrophobic and hydrophilic balance between two segments and the hydrophobic interactions/hydrogen bonding adopted by the second blocks affected the formation of hydrogels. The polypeptide topology and composition were the two main factors to affect the non-covalent interactions, which are hydrogen bonding and hydrophobic interactions, and the amphiphilic nature exhibited by these diblock copolypeptides. We found that linear PLL-*b*-PPhe diblock copolypeptides with DP lower than 30 could not form hydrogels in an aqueous solution at the polypeptide concentration below 10 wt%. This study highlighted that star-shaped topology could facilitate the hydrogelation of *s*-PLL-*b*-PY diblock copolypeptides due to the multiple interacting depots between PY segments. It can be seen that the 6-armed diblock copolypeptides had greater hydrogelation ability, as compared to 3-armed ones, which is due to the fact that each 6-armed polypeptide chain has more inter-chain interacting depots than the 3-armed one. *s*-PLL-*b*-PPhe exhibited lower CGC than *s*-PLL-*b*-PVal and *s*-PLL-*b*-PLeu due to the additional π–π interactions between benzyl groups on the PPhe segment and cation–π interactions between PLL and PPhe segments. Moreover, it is obvious that all the hydrogel samples could form 3D nano-assemblies by self-assembly and adopt different conformations from each segment. The difference of morphology between *s*-PLL-*b*-PPhe and the others, between 3-armed and 6-armed samples of PPhe, demonstrated the effects of hydrophobic component composition and arm number on the structure of polymers. Our previous study showed that CGCs of 3*s*-PLL_20_-*g*-PPhe_0.3_ and 6*s*-PLL_20_-*g*-PPhe_0.3_ graft copolypeptides were 5.0 and 7.0 wt%, respectively [28,32], which were higher than those of the corresponding *s*-PLL-*b*-PPhe with the same arm number in this study. Obviously, the hydrogelation of *s*-PLL-*b*-PPhe diblock copolypeptides is more effective than the graft copolopeptides, which could be attributed to the efficient packing of *s*-PLL-*b*-PPhe. Consequently, *s*-PLL-*b*-PPhe could self-assemble to form interconnected, membraneous networks to accommodate water molecules at relatively lower polypeptide concentrations.

## 3. Conclusions

We demonstrate that star-shaped *s*-PLL-*b*-PPhe, *s*-PLL-*b*-PVal, and *s*-PLL-*b*-PLeu diblock copolypeptides but not PLL-*b*-PAla can self-assemble to form hydrogels with CGCs ranged between 1.0 to 7.0 wt%, depending on the arm number and hydrophobic, sheet-like polypeptide segment. The amphiphilic balance between polypeptide blocks and the hydrogen bonding/hydrophobic interactions exerted by the hydrophobic polypeptide segment dictated the hydrogelation and mechanical properties of the hydrogels. Based on the different non-covalent interactions (hydrophobic, hydrophilic and, hydrogen bonding), the formation of multiple interacting depots could facilitate hydrogelation at low polypeptide concentrations. The experimental data showed that the CGCs of 6-armed diblock copolypeptides were lower than those of 3-armed ones and 6*s*-PLL_21_-*b*-PPhe_5.3_ exhibited the lowest CGC value (1.0 wt%) among all. The additional π–π and cation–π interactions rendered the *s*-PLL-*b*-PPhe exhibiting better hydrogelation ability than *s*-PLL-*b*-PVal and *s*-PLL-*b*-PLeu. This study clearly illustrated that the combination of a sheet-like segment as a hydrogelator and star-shaped topology can trigger the hydrogelation of these block copolypeptides.

## 4. Materials and Methods

### 4.1. Materials

Schlenk-line techniques were used to handle all compounds, which are sensitive to air and moisture under a nitrogen atmosphere. All chemicals and solvents were ACS reagent grade and used without any purification process unless otherwise noted. *N,N*-dimethylformamide (DMF, Macron Fine Chemicals™, Radnor, PA, USA), hexane (ECHO CHEMICAL CO., LTD, Miaoli County, Taiwan), and THF (J.T.Baker, Phillipsburg, NJ, USA) were dried by 4A molecular sieves (UniRegion Bio-Tech Co., Hsinchu, Taiwan), calcium hydride (90–95%, Alfa Aesar, Shanghai, China), and sodium metal (99.95%, in mineral oil, Sigma-Aldrich, St. Quentin Fallavier, France), respectively. Z-L-lysine (ZLL) (Sigma-Aldrich, Schaffhausen, Switzerland), L-leucine (Leu) (Sigma-Aldrich, Tokyo, Japan), L-phenylalanine (Phe) (Sigma-Aldrich, Tokyo, Japan), L-alanine (Ala) (Sigma-Aldrich, Taufkirchen, Germany), and L-valine (Val) (Sigma-Aldrich, St. Louis, MO, USA) N-carboxyanhydrides (NCAs) were prepared based on a procedure reported in the previous papers [28,32]. 

### 4.2. Synthesis of Star-Shaped Poly(_L_-lysine)-Based Block Copolypeptides

Following the reported procedure, an initiator was used for the synthesis process of the diblock copolypeptides to sequentially polymerize ZLL and other NCAs [28,32,49]. The block ratio was set to be 4:1. The feed molar ratio of 3-armed initiator (1,1,1-tris(hydroxymethyl)propane) (Acros Organics, Schwerte, Germany) to ZLL NCA was set to be 1:60. The feed molar ratio of 6-armed initiator (dipentaerythritol) (Acros Organics, Shanghai, China) to ZLL NCA was set to be 1:120. A detailed procedure for the synthesis of 3-armed-poly(Z-L-Lysine)_20_-*block*-poly(_L_-phenylalanine)_5_ (3*s*-PZLL_20_-*b*-PPhe_5_) is given below as an example.

In a glove box, ZLL (2.0 g) and 3-armed initiator (14.6 mg) were dissolved in anhydrous DMF (6.54 mL and 5.45 mL, respectively) in round bottom flasks. A TMG stock solution (6.0 mM, 24.62 μL, Aldrich, China) was dropped into the initiator solution. The solution was heated and stirred until becoming transparent. Then, the ZLL NCA solution was added to the completely dissolved initiator solution. The mixture was stirred outside the glove box at RT under a nitrogen atmosphere for 48 h. Phe NCA (312.4 mg) was dissolved in anhydrous DMF (1.64 mL) and poured into the reaction mixture. Upon stirring for an additional 24 h, the reaction mixture was dialyzed against DMF, methanol (ECHO, Taiwan), and DI water for 2 h, 24 h, and 72 h, respectively, using cellulose membrane tubes (MWCO 6000–8000 g mol^−1^, Spectrum Laboratories, USA). A white product was collected after the lyophilization of the dialyzed solution (yield: 80–90%).

The deprotecting reaction was conducted by using hydrobromic acid (HBr) (Acros, Israel) to remove the Z group. As a reference procedure, the polypeptide (1.5 g) was completely dissolved in trifluoroacetic acid (TFA, 75 mL, Alfa Aesar, Lancashire, UK) in a round bottom flask. A 5-fold excess with respect to the Z group molar ratio of a 33 wt% HBr solution in acetic acid was added to the solution slowly [16,28,31,32]. The reaction mixture was precipitated into diethyl ether (ECHO, Taiwan) after stirring for 30 min at RT. The precipitate was collected by using centrifugation and washed twice with diethyl ether. After drying under a vacuum, DI water was used to dissolve the collected product. Then, the solution was poured into dialysis membranes (MWCO 6000–8000 g mol^−1^) and dialyzed against DI water for 72 h. A white spongy product was collected after freeze-drying (yield: 90–95%).

### 4.3. Characterization of Block Polypeptides

The polypeptides were characterized by proton nuclear magnetic resonance (^1^H-NMR) and gel permeation chromatography–light scattering (GPC-LS). *s*-PZLL dissolved in TFA-*d_1_* and *s*-PLL-*b*-PY dissolved in DMSO-*d_6_* were analyzed by using a BRUKER ADVANCE III HD NMR (600 MHz, Bruker Corporation, Karlsruhe, Germany). The 3-armed and 6-armed PZLL homopolypeptides were analyzed by a Viscotek GPS-LS system (Malvern Instruments Limited, Worcestershire, UK) equipped with two ViscoGEL I-Series columns (catalog number: I-MBHMW-3078 and I-MBLMW-3078, Viscotek) and three detectors, which are Dual 270 viscometer, right-angle light scattering, and VE3580 RI for efficient separation. That GPS-LS system was performed at 55 °C and 1.0 mL min^−1^ of flow rate on the star-shaped PZLL homopolypeptides. The eluent was DMF containing 0.1 M LiBr (Alfa Aesar, Shanghai, China), and the calculation standard was polystyrene (molecular weight: 25,000 g mol^−1^, Alfa Aesar, Shanghai, China). The 3-armed and 6-armed PZLL homopolypeptides were dissolved in DMF completely. Before GPC-LS analysis, those solutions were passed through a 0.2 μm PTFE filter (13 mm, Finetech, Taiwan). The equipped software (OmniSEC 4., Viscotek, Malvern Instruments Limited, Worcestershire, UK) was employed to determine their number-average molecular weights (*M_n_*) and molecular weight distributions (*M_w_*/*M_n_*).

### 4.4. Preparation Polypeptide Hydrogels and Determination of Critical Gelation Concentration (CGC)

At RT, the freeze-dried diblock copolypeptide samples were completely dissolved in DI water by using the ultrasound and vortex mixer to obtain clear solutions. Then, the resultant mixtures were placed on a counter overnight at RT to reach equilibrium. The vial inverting method was employed for determining the CGC of the samples [26,50,51]. Following the above procedure, the hydrogels were prepared with provided concentrations in 5 mL vials. The CGCs were determined as the samples did not flow for 60 s after the vials were inverted.

### 4.5. Characterization of Polypeptide Secondary Conformation

Circular dichroism (CD), X-ray diffraction (XRD), small-angle X-ray scattering (SAXS), and Fourier transform infrared (FTIR) were employed to measure the polypeptide secondary conformation and self-assembled structures. CD measurements of polypeptide samples were conducted at the concentration of 0.1 mg mL^−1^ on a JASCO J-815 spectrometer (JASCO Corporation, Japan) in a 0.1 mm quartz cell from 190 nm to 260 nm of the wavelength. A Rigaku Ultima IV-9407F701 X-ray spectrometer (Rigaku Corporation, Japan) was used to record XRD patterns of star-shaped polypeptide hydrogel samples. XRD system scanned the patterns from 2θ = 5° to 40° at a speed of 10° min^−1^ by using radiation (50 kV, 250 mA) and Cu K alpha (0.154 nm). At 25 °C, SAXS profiles of the sol and gel samples were obtained under 4 × 10^−1^ torr and the voltage and current were controlled in 45 kV and 650 μA, respectively. The SAXS measurements were run on a Bruker diffractometer (NanoSTAR-SHAPED U system, Bruker AXS GmbH, Germany). The samples were contained in 1 mm quartz capillary tubes at different sol and gel concentrations. Before the measurements, a silver behenate was used as a standard sample for calibrating the SAXS. Attenuated total reflectance FTIR spectra of freeze-dried gel samples were recorded on a Thermo Nicolet Nexus 670 FTIR spectrometer (Thermo Electron Corporation, USA).

### 4.6. Characterization of Polypeptide Hydrogels

A Hitachi SU8010 microscope (Hitachi High-Technologies Corporation, Japan) was used to take the field emission scanning electron microscopy (FE-SEM) images of freeze-dried samples. For rheological measurements, the s-PLL-*b*-PY polypeptide hydrogel samples were prepared at 5.0 wt% and 8.0 wt% of polypeptide concentration and tested at RT with a 25 mm diameter aluminum plate geometry. By varying angular frequency and strain amplitude, a DISCOVERY HR-2 (TA Instruments/Waters Corporation, USA) controlled strain rheometer was used to measure the dynamic variety on loss modulus (G′′) and storage modulus (G′) of the polypeptide hydrogel samples. To perform the hydrogel recovery measurements, the polypeptide hydrogel structures were broken down by these steps: nonlinear, large-amplitude oscillations (100% of strain) at an angular frequency of 1.0 rad s^−1^ for 300 s, followed by monitoring the mechanical strength recovery of hydrogels at the same angular frequency with a constantly low strain of 1%.

## Data Availability

The data presented in this study are available on request from the corresponding author.

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
