# Peer review of "Polypeptide Composition and Topology Affect Hydrogelation of Star-Shaped Poly(_L_-lysine)-Based Amphiphilic Copolypeptides"

_gels, 2021, doi:10.3390/gels7030131_

Round 1
Reviewer 1 Report
The manuscript by Thi Ha My Phan et al., Polypeptide composition and topology affect hydrogelation of star-shaped poly(L-lysine)-based amphiphilic copolypeptides provides design, chemical synthesis and characterization of polypeptide hydrogels. Overall, the data are interesting, are presented clearly and the manuscript is well written. My comments and questions to the authors are as follows:
#1. page 6, lines 175 172-176: It should be compared at the same concentration wt%, but was there a difference in porous structure among 6 types of gels? Especially, readers would be interested to know if there is any difference among PPhe-variant, PVal, and PLeu.
#2. page 6, lines 196-201 & page 11 lines378-382: Are these gels self-healing gels? In figures 4 & 5, the gels were broken at 100 sec at 100% strain, but the G' values are recovered after 400 sec. The authors were encouraged to explain more in both result/discussion section and methods section.
#3. Side chain of Lys-NH2 and aromatic ring interactions are well known as cation-pi interaction. The authors seem to emphasize the contribution of pi-pi interaction, but is it necessary to consider the contribution of cation-pi interaction?
Author Response
The manuscript by Thi Ha My Phan et al., Polypeptide composition and topology affect hydrogelation of star-shaped poly(L-lysine)-based amphiphilic copolypeptides provides design, chemical synthesis and characterization of polypeptide hydrogels. Overall, the data are interesting, are presented clearly and the manuscript is well written.
Response: The authors thank the reviewer’s comments above and revised our manuscript according to the specific comments.
My comments and questions to the authors are as follows:
#1. page 6, lines 175 172-176: It should be compared at the same concentration wt%, but was there a difference in porous structure among 6 types of gels? Especially, readers would be interested to know if there is any difference among PPhe-variant, PVal, and PLeu.
Response: The authors thank the reviewer’s comment. We chose the concentration based on the CGCs for the samples. That was the reason we chose 5 wt% for 4 samples having CGCs lower than 5wt% and 8 wt% for 2 remaining ones having the CGCs higher than 5 wt%.
“At the same concentration (5wt%), 6s-PLL21-b-PPhe5.3 had a thin elongated fibrillar structure whilst 6-armed samples of PLeu and PVal had membranous networks (Figure 3b, d, f). Moreover, unlike 6s-PLL21-b-PPhe6.3 sample, 3s-PLL22-b-PPhe6.3 showed membranous morphology.”
We have included the above information in the revised manuscript as highlighted in lines 185-188 of page 7.
#2. page 6, lines 196-201 & page 11 lines378-382: Are these gels self-healing gels? In figures 4 & 5, the gels were broken at 100 sec at 100% strain, but the G' values are recovered after 400 sec. The authors were encouraged to explain more in both result/discussion section and methods section.
Response: The authors thank the reviewer’s comment. The recovery test showed that the hydrogels were broken at 100 sec at 100% strain, but the G' values are recovered after 400 sec, revealing that the hydrogels are recoverable. It also suggests that the hydrogels might be injectable due to their ability to recover from sol to gel. In the literature, the self-healing ability of hydrogels was tested by cutting them into two halves and see whether they can be reconnected or not.
#3. Side chain of Lys-NH2 and aromatic ring interactions are well known as cation-pi interaction. The authors seem to emphasize the contribution of pi-pi interaction, but is it necessary to consider the contribution of cation-pi interaction?
Response: The authors thank the reviewer’s comment. The authors also think that cation-pi interaction also plays a role in hydrogelation.
We have included the above information in the revised manuscript as highlighted in lines 24-25 of page 1, 144-146 of page 5, 276-278 and 301-303 of page 10.

Reviewer 2 Report
Review – Gels- 210812
The manuscript by Hu, Jan, and coworkers describes the synthesis of 3- and 6-arm polypeptide stars as gelators. After polymerizing lysine and a hydrophobic amino acid N-carboxyanhydride from 3- or 6-hydroxyl group-containing cores, the authors study the gelation by rheology, microscopy, and scattering. They find 6-arm polymers to facilitate gelation at lower concentrations than the 3-arm polymers, and that phenylalanine groups capable of pi-pi stacking interactions promote gelation at lower concentrations than polymers with polyleucine, polyvaline, or polyalanine blocks. Since hydroxyl groups are not as nucleophilic as amines, the authors use tetramethyl guanidine to facilitate polymerization. While the authors report SEC data in tables, it would be helpful to see these traces to verify that the second peptides indeed add exclusively to the stars rather than homopolymerizing, for example, initiated by water. The manuscript would also benefit from a more thorough discussion of their results in the context of the literature on other star polymer-based hydrogelators. Specific comments are provided below. The structure-function relationships connecting arm number and hydrophobic component composition to hydrogel properties will certainly interest the readership of gels, but more thorough molecular characterization (i.e., SEC) and associated discussion is needed to confirm the structure of these gelators with complex architectures.
- In Table S1, the samples are named ZLL20, suggesting the degree of polymerization in all cases is 20. However, it seems to be the case that the data in this table are for polymerization of lysine blocks with different lengths.
- In Figure S2, why is the water peak downshifted relative to the water peaks in the other spectra? Also, these peaks are labeled as just ‘a’ and ‘b’ protons from the core, but the likely contain a significant fraction of water, since these initiator/core protons are a small fraction of the star polymers.
- In the caption of Figure S3, specify what solvent / buffer was used for these measurements.
- To confirm successful preparation of the star polymers, size exclusion chromatography (SEC) traces are needed in addition to the Mn and dispersity values provided in the tables. Particularly since hydroxyl groups are less nucleophilic than typical amine initiators for polypeptide synthesis, these chains may not all contain two blocks, for example, and the author’s prior publication on these similar star polymers contains SEC traces of just the first polylysine block. SEC data, particularly when combined with light scattering, will help to establish chain extension of PLL chains on each arm with a Y block rather than separate homopolymerization of the Y block. Similarly, the SEC characterization should be described in the main text together with the NMR-based structural characterization.
- In the introduction, it would be useful to explicitly discuss what others have found by varying the arm number and arm length of polypeptide hydrogels. This will help to establish the novelty of this work, and allow discussion of these results in the context of other hydrogelating star polymers.
- The authors suggest that the reason for the ‘shear thickening’ seen in the loss moduli is still debated, however wouldn’t the increase in loss modulus as the storage modulus decreases at high strain merely reflect the increasingly viscous, liquid-like character of the gel?
- The microscopy images and rheological data suggest the formation of gels, whereas the x-ray data discussion seems to imply the formation of fibrils and sheets – additional explanation would be helpful.
Author Response
The manuscript by Hu, Jan, and coworkers describes the synthesis of 3- and 6-arm polypeptide stars as gelators. After polymerizing lysine and a hydrophobic amino acid N-carboxyanhydride from 3- or 6-hydroxyl group-containing cores, the authors study the gelation by rheology, microscopy, and scattering. They find 6-arm polymers to facilitate gelation at lower concentrations than the 3-arm polymers, and that phenylalanine groups capable of pi-pi stacking interactions promote gelation at lower concentrations than polymers with polyleucine, polyvaline, or polyalanine blocks. Since hydroxyl groups are not as nucleophilic as amines, the authors use tetramethyl guanidine to facilitate polymerization. While the authors report SEC data in tables, it would be helpful to see these traces to verify that the second peptides indeed add exclusively to the stars rather than homopolymerizing, for example, initiated by water. The manuscript would also benefit from a more thorough discussion of their results in the context of the literature on other star polymer-based hydrogelators.
Response: The authors thank the reviewer’s comments above and revised our manuscript according to the specific comments.
Specific comments are provided below.
The structure-function relationships connecting arm number and hydrophobic component composition to hydrogel properties will certainly interest the readership of gels, but more thorough molecular characterization (i.e., SEC) and associated discussion is needed to confirm the structure of these gelators with complex architectures.
Response: The authors thank the reviewer’s comment. We have added a brief discussion the structure-function relationships connecting arm number and hydrophobic component composition to hydrogel properties as shown in the following.
It can be seen that the 6-armed diblock copolypeptides had greater hydrogelation ability as compared to 3-armed ones, which is due to the fact that each 6-armed polypeptide chain has more inter-chain interacting depots than the 3-armed one. s-PLL-b-PPhe exhibited lower CGC than s-PLL-b-PVal and s-PLL-b-PLeu due to the additional p-p interactions between benzyl groups on PPhe segment and cation-p interactions between PLL and PPhe segments. Moreover, it is obvious that all the hydrogel samples could form 3D nano-assemblies by self-assembly and adopt different conformations from each segment. The difference of morphology between s-PLL-b-PPhe and the others, between 3-armed and 6-armed samples of PPhe demonstrated the effects of hydrophobic component composition and arm number on the structure of polymers. Our previous study showed that CGCs of 3s-PLL20-g-PPhe0.3 and 6s-PLL20-g-PPhe0.3 graft copolypeptides were 5.0 and 7.0 wt%, respectively [28, 32], which were higher than those of the corresponding s-PLL-b-PPhe with the same arm number in this study. Obviously, the hydrogelation of s-PLL-b-PPhe diblock copolypeptides is more effective than the graft copolopeptides, which could be attributed to the efficient packing of s-PLL-b-PPhe. Consequently, s-PLL-b-PPhe could self-assemble to form interconnected, membraneous networks to accommodate water molecules at relatively lower polypeptide concentration.
We have included the above information in the revised manuscript as highlighted in lines 273-289 of page 10.
In Table S1, the samples are named ZLL20, suggesting the degree of polymerization in all cases is 20. However, it seems to be the case that the data in this table are for polymerization of lysine blocks with different lengths.
Response: The authors thank the reviewer’s comment. We have corrected the 3s-PZLL20 to 3s-PZLL22, 6s-PZLL20 to 6s-PZLL21 and the DP of second segments based on the new DP of PZLL in our manuscript.
In Figure S2, why is the water peak downshifted relative to the water peaks in the other spectra? Also, these peaks are labeled as just ‘a’ and ‘b’ protons from the core, but the likely contain a significant fraction of water, since these initiator/core protons are a small fraction of the star polymers.
Response: The authors thank the reviewer’s comment. Based on this comment, the author thinks the review was asking about the NMR spectra in Figure S1 on page S5.
Based on 1H NMR, only the spectra for 3-armed and 6-armed PLL-b PLeu samples showed we the downshift of the water peak. The reason is unknown. We agree with it and have changed the label to H2O peak as shown in Figure S1.
In the caption of Figure S3, specify what solvent / buffer was used for these measurements.
Response: The authors thank the reviewer’s comment. Deionized (DI) water was used for these measurements. We have added this information in the caption of Figure S3 on page S7.
To confirm successful preparation of the star polymers, size exclusion chromatography (SEC) traces are needed in addition to the Mn and dispersity values provided in the tables. Particularly since hydroxyl groups are less nucleophilic than typical amine initiators for polypeptide synthesis, these chains may not all contain two blocks, for example, and the author’s prior publication on these similar star polymers contains SEC traces of just the first polylysine block. SEC data, particularly when combined with light scattering, will help to establish chain extension of PLL chains on each arm with a Y block rather than separate homopolymerization of the Y block. Similarly, the SEC characterization should be described in the main text together with the NMR-based structural characterization.
Response: The authors thank the reviewer’s comment. We would like to acquire the SEC traces of these star-shaped block copolypeptides and have tried very hard to find suitable organic solvents to dissolve these star-shaped block copolypeptides for GPC-LS analysis. However, they can only dissolve in organic acids, which could not be the mobile phase in GPC-LS. Based on Heteronuclear single quantum coherence (HSQC) 1H and 13C NMR spectra published in our previous study [Tang, C.-C.; Zhang, S.-H.; My Phan, T. H.; Tseng, Y.-C.; Jan, J.-S. Polymer 2021, 123891.], the synthesis of the star-shaped polypeptides was successful, as seen by the disapperance of the protons on the intiators (–CCH2OH–) at 66.3 ppm and the appearance of the protons on the s-PZLL (–CCH2OC(O)–) at 40–43 ppm. It demonstrated that all OH groups of the initiators were completely initiated by PZLL block. Hence, we believe that Y segment would continue to extend from PZLL segment on each arm. Moreover, these star-shaped block copolypeptides are very soluble in DI water, suggesting that there should be no individual Y segment in the samples. The Y segments are hydrophobic and cannot be dissolved in water.
In the introduction, it would be useful to explicitly discuss what others have found by varying the arm number and arm length of polypeptide hydrogels. This will help to establish the novelty of this work, and allow discussion of these results in the context of other hydrogelating star polymers.
Response: The authors thank the reviewer’s comment. We have added a brief discussion about this aspect in lines 84-87 of page 2.
In previous papers, the researchers reported that the increase of block copolypeptides arm length and arm number led to decreasing in critical gelation concentration (CGC) and increasing in hydrogelation of polymers [29, 31] while it had no obvious trend for graft copolypeptides [28].
The authors suggest that the reason for the ‘shear thickening’ seen in the loss moduli is still debated, however wouldn’t the increase in loss modulus as the storage modulus decreases at high strain merely reflect the increasingly viscous, liquid-like character of the gel?
Response: The authors thank the reviewer’s comment. Based on the references 16 and 28 cited in our manuscript, it has many reasons for the “shear thickening” but they are still assumptions. We thought that the reviewer’s explanation might be possible.
We have included the above information in the revised manuscript as highlighted in lines 198-200 of page 7.
The microscopy images and rheological data suggest the formation of gels, whereas the x-ray data discussion seems to imply the formation of fibrils and sheets – additional explanation would be helpful.
Response: The authors thank the reviewer’s comment. Those analyses were used for different purposes because they showed the result about the structures with dimensions at different scales. The SEM and rheology were employed to analyze the gel matrix and strength properties of hydrogels at microscale, while SAXS was used to analyze the self-assembled structure at nanoscale.
